# Game Theoretic Solution for Power Management in IoT-Based Wireless Sensor Networks

**DOI:** 10.3390/s19183835

**Published:** 2019-09-05

**Authors:** Muhammad Sohail, Shafiullah Khan, Rashid Ahmad, Dhananjay Singh, Jaime Lloret

**Affiliations:** 1Institute of Computing, Kohat University of Science and Technology, Kohat 26000, Pakistan (M.S.) (S.K.); 2Department of Physics, Kohat University of Science and Technology, Kohat 26000, Pakistan; 3Department of Electronics Engineering, Hankuk University of Foreign Studies, Yongin 17035, Korea; 4Universitat Politecnica de Valencia, C/Paranimf, 1, Grao de Gandia, Gandia, 46370 Valencia, Spain

**Keywords:** evolutionary game, energy efficiency, game theory, wireless sensor networks

## Abstract

Internet of things (IoT) is a very important research area, having many applications such as smart cities, intelligent transportation system, tracing, and smart homes. The underlying technology for IoT are wireless sensor networks (WSN). The selection of cluster head (CH) is significant as a part of the WSN’s optimization in the context of energy consumption. In WSNs, the nodes operate on a very limited energy source, therefore, the routing protocols designed must meet the optimal utilization of energy consumption in such networks. Evolutionary games can be designed to meet this aspect by providing an adequately efficient CH selection mechanism. In such types of mechanisms, the network nodes are considered intelligent and independent to select their own strategies. However, the existing mechanisms do not consider a combination of many possible parameters associated with the smart nodes in WSNs, such as remaining energy, selfishness, hop-level, density, and degree of connectivity. In our work, we designed an evolutionary game-based approach for CH selection, combined with some vital parameters associated with sensor nodes and the entire networks. The nodes are assumed to be smart, therefore, the aspect of being selfish is also addressed in this work. The simulation results indicate that our work performs much better than typical evolutionary game-based approaches.

## 1. Introduction

In this digital era, every object is getting smart, i.e., intelligence is embedded for quick and rational decisions. Internet of things (IoT) and next-generation networks are playing vital roles in connecting, operating, and monitoring distinct objects for smart decision making [1]. The enabling underlying technology for IoT are wireless sensor networks (WSN). WSN consists of self-configuring sensor nodes capable of sensing, collecting, and forwarding data to the sink node, known as base station (BS). The sensor nodes combine sensing, computing, and communication in a single unit. The data reach to the BS through intermediate nodes called relay nodes which have routing capability. We can use WSNs in many applications, like real-time tracking, environmental monitoring, disastrous regions, industrial operations, and health care [2]. The sensor nodes are deployed in remote and hostile regions, working continuously by discharging their only sources of energy, i.e., batteries. The embedded batteries of sensor nodes cannot be recharged or replaced. It is important to mention that some of the built-in power technologies helped to improve the lifetime of the WSNs. However, nodes are deployed in tough environments for a very long time, therefore, it is necessary to use the power source efficiently, and this can be only achieved through an efficient power management scheme [3]. We know that sensor nodes do sensing, processing, and communication. They take maximum power consumption in communication [4], therefore, researchers focus on communication by establishing some better schemes that ensure efficient energy utilization.

Different energy management schemes are used to extend the lifetime of the networks such as workload management, mobile networks, clustering, optimal deployment of nodes, intelligent data gathering, and energy harvesting from the environment [5].

Some of the selfish sensor nodes drain out energy by communication, mostly keeping off their antennas. Therefore, we need some powerful tool that can evaluate and make the decisions rationally. One of the candidate schemes to optimize energy consumption is clustering. Clustering techniques are mainly focused on the communication part of the WSNs by prolonging the lifetime. It has been involved in many applications of the WSNs as well [6]. The election of the cluster head (CH) is an important issue in the overall process. In the CH selection process, many parameters are considered keeping in view the requirements and scenarios of WSNs. In this process, every node cannot be suitable to assume the role of CH due to low energy, unimportant location, and distance from BS, etc. To select the proper node as a CH, game theory (GT) has been found very suitable to make rational decisions [7]. GT is a powerful mathematical tool used to analyze and predict the decisions of individuals under conflict situations.

An evolutionary game theoretic model can be used to efficiently divide the WSNs into various clusters in such a way that each node can make interactive rational decisions for their own benefits. Such mechanisms can guarantee the tradeoff between the rationality of nodes with the most efficient utilization of energy in the entire network. In evolutionary games, the nodes, referred to as players, rationally adopt a strategy set, which continuously varies with the passage of time by reading the behavior of other nodes in the network. The CH can be optimally selected by using this game model in an iterative fashion. The network evolves with the passage of time and learns the best preferences, which are ultimately used for CH selection in an efficient way. This theory is originated from the area of population dynamics, where nodes intelligently adopt the most suitable strategies to maximize their payoff [8].

This work proposes a technique for energy-efficient clustering by combing an evolutionary game with some novel additional improvements. The main objective of this work is energy optimization by efficient CH selection and evaluation process of all the nodes. Three main sets of nodes are determined by the BS, i.e., (a) suitable, (b) not suitable, and (c) prohibited nodes for being CHs. The nodes’ performance is continuously assessed by the BS to determine these sets. The selfish nodes are marked prohibited in the beginning. The CH selection is then further made dependent on some key parameters considered by the BS. The BS predicts these parameters’ values by itself to reduce the communication overhead on the nodes. A new concept of nodes’ importance based on the closed neighbors (CNs) is also introduced in this work. Once the parameters are evaluated, the BS gives benefits to the most appropriate nodes or gives penalties to the non-suitable nodes. An evolutionary game is applied to all of these nodes combined in suitable and non-suitable sets, along with their profits and penalties values.

The rest of the paper is organized as follows. Section 2 comprises of related work; many related mechanisms used for energy optimization have been discussed. Section 3 explains the detail of our proposed mechanism. Many subsections are used to clearly explain the entire proposed mechanism. Section 4 consists of the experimental results where two clusters-based and two evolutionary clusters-based approaches were taken and compared with our proposed mechanism. Lastly, conclusion and future works are given in Section 5.

## 2. Related Work

Game theory (GT) is a mathematical construct that defines the conditions of cooperation, non-cooperation, and repetition among rationally independent decision-makers. In the recent literature, this theory has been proven by many researchers as one of the best tools in wireless ad hoc networks for various purposes. GT can be used in WSNs for a variety of purposes including security, clustering, routing, load balancing, quality of service (QoS), power control, intrusion detection, selfish node management, and efficient resource management [9].

Energy management in WSNs through clustering techniques and GT have been considered for many years. This has opened a rich area for researchers in the field of WSNs. However, there are some open problems to be settled. We briefly summarize some of the major contributions in the field. It is important to mention that some works have focused only on the clustering process and have improved the lifetime of WSNs successfully. On the other hand, others focused on the cluster size with taking two parameters for measuring the lifetime of a sensor node, namely residual energy and cluster size. Clustering has been also used in area coverage awareness using residual energy and location [4].

Clustering has two types, namely centralized and distributed. In the first type, the BS is responsible for all types of cluster formations, while in distributed clustering the hierarchical structure with low energy models is generally used to form clusters and select CHs. The drawback of centralized clustering is due to its centralized implementation and complex design. This is not quite useful for sensor networks containing a huge number of nodes. Moreover, this mechanism is dependent upon the position of nodes and CHs, and respective cluster distribution may not be equal.

The data fusion technique for distributed clustering is more energy-efficient protocol. This has decreased the importance of CHs and cluster creation by centralized stations. The data fusion models use low energy and hierarchical structure for cluster formation and CHs selection.

The ICCBP (inter-cluster chain-based protocol) is a clustering algorithm that utilizes multi-hop and intra-cluster communication to replace CHs when the existing CHs run out of their energy [10]. The hop distance and residual energy are the CH selection parameters. Similarly, in Ref. [11] an energy-efficient multi-hop routing is controlled using clustering for WSNs with the help of two parameters, namely residual energy and distance.

We found in the literature a new approach called cognitive radio WSN (CR-WSN) [12,13,14,15]. CR-WSN has received significant attention. This scheme exploits the spectrum resource efficiently for huge traffic. The mechanism is capable of reducing packet loss and power waste. It can also manage a high degree of buffer and communication quality. This approach enhances the spectrum utilization, network efficiency, life time of WSNs, and the end-to-end goal is achieved.

On the other hand, it is well known that power efficiency is increased by having distant but densely populated hops. This fact can help in power conservation in CR-WSNs. However, WSN clustering methods cannot be implemented directly CR-WSNs. The reason is communication-based decision making for CH selection. In the conclusion, CR-WSN is still an open area of research that needs due attention. However, game theoretic approaches for spectrum utilization, packet arrival prediction, energy-efficient routing, media access control (MAC) protocols, and design of cross-layer algorithms for better power efficiency have already been introduced. Clustering is simple and improved scalability suitable for WSN has the advantage of low energy consumption and is good in many energy-efficient routing algorithms like LEACH, LEACH-C [16], and PEGASIS [17]. Researchers in [18] tried to solve the energy conservation problem but they made many assumptions in their approaches like nodes from clusters should have a lot of information about their neighbor, also they ignored the selfishness of the nodes too.

The selection/election of CH in the cluster is a very important task. Appropriate CH election is an essential consideration and, as mentioned earlier, location and connectivity are two of the most important parameters for such selection. Other techniques do exist in the literature, like in ref. [19] a fuzzy logic technique was used, considering neighbor nodes and residual energy parameters. In ref. [20], the trajectory-based clustering method for CH selection was introduced and in ref. [21], relative density parameter was used to improve LEACH.

GT has been increasingly used in WSNs for a variety of issues, like efficient energy usage, more control over power transmission, data collection, and communication security. More specifically, GT has been very successful in the design schemes of WSNs.

Normally, the nodes in WSNs act selfishly by conserving their residual energy by denying the receiving information from associates. This mostly happens in the multi-hop network during transmission. To get the optimum values for energy efficiency it the noncooperative game theoretic approach was used, like [22] where authors used the same approach for the election of the CHs in clustering. A node calculates the optimal probability by itself and declares itself as a CH or not, depending upon the maximization of payoff.

GT together with the concept of the hierarchical topology is used in some algorithms. In ref. [23], authors have developed a game-theoretic scheme considering residual energy and average energy loss for maximizing payoff from CH selection. Similarly, in ref. [24] coalitional GT was adopted to determine the most efficient route in a group, considering the power consumption. In other works such as [25], routing in WSNs was done using Bayesian game, where Harsanyi transformation was used to create a static game of complete but imperfect information. A topology control algorithm in [26], based on the ordinal potential game, was proposed by designing a payoff function that considers two parameters only, which are network connectivity and the energy balance of nodes.

In ref. [27], evolutionary game theory (EGT) was found to be affected by the frequency of the competing strategies in the population in the communication networks.

It was noticed that the authors in [28] considered an evolutionary game approach for duty cycle and fuzzy logic for routing, but with only two parameters—residual energy and distance. While on another hand, double time Nash equilibrium was counted for suitable CH selection in [29] but they considered only parameter residual energy.

An evolutionary game model “Game theory based Energy Efficient Clustering routing protocol (GEEC)” [30] was proposed to extend the network lifetime by achieving the energy exhaust equilibrium. The nodes were divided into two categories of candidates for CH or cluster member (CM). A game was designed to illustrate the possibilities for both categories for being CHs. Some incentives, in the form of profits and penalties, were designed to push the nodes toward their willingness for being CHs. The work is well-presented and all the possible aspects associated with the game processing are presented. Moreover, the proposed mechanism was compared with LEACH and LEACH-C. The results claim that GEEC gave better results than the experimented protocols.

In another article [31], the evolutionary game was applied to calculate the traffic amount among different regions in the network. The network clusters were further divided into subproblems. The work inherited the key characteristics of GT like distributed decision, dynamic network, and resource usage preferences. The area associated with each sender node was segmented into various subregions. The evolutionary game was then applied for the selection of these subregions while considering the energy levels of the involved nodes. During the communication, it was also assured that all the involved nodes consume their energies in an equal fashion by applying classical GT. The main theme of the work was to combine the evolutionary game with the classical GT. The work was compared with some other protocols in various directions. The best results among all the experimented protocols were claimed by the authors at the end of work.

Although some of the abovementioned algorithms and protocols based on game theory can achieve network topology control and improve network performance, they cannot guarantee the connectivity and robustness of the network. Additionally, the parameters like residual energy, degree of connectivity, distance, and energy efficiency of the nodes are not fully and accurately considered. Nor are these parameters taken combined and rationalized as a whole. Therefore, we need a mechanism to consider the proper selection of CH by considering appropriate parameters rationally, so that we can manage the power efficiently. Our proposed scheme contributed better results, as given in Section 4. For a more clear view of the related work, the following Table 1 is summarized with some key features, advantages, and disadvantages/research gaps.

## 3. The Proposed Mechanism

We know that game theory is the study of mathematical models of planned interaction between rational decision-makers. We found its application in almost all areas of social sciences, as well as in computer science too. Especially in the design of WSNs, game theory has been proven very suitable. More precisely, the EGT encompasses a dynamic process and the replicator dynamics [32]. However, a model based on the EGT considers the devising of a traditional game by using population. Individual players can be of any number in such model and they can adopt strategies against other players in the same population. The EGT has now been extensively used and its applications are to predict the variable tendency of a population’s behavior.

The proposed work uses an updated evolutionary game model for energy optimization and an efficient CH selection process in cluster-based WSNs. The main theme of the evolutionary game in this work is inspired by GEEC [30]. The work assumes that all the nodes are intelligent and they can perform according to their preferences. However, the system has the authority to mark the nodes according to their performance before considering them for taking the responsibility of cluster-headship. Our work is divided into various segments. In Section 3.1, we discuss the main assumptions and the fundaments associated with this work. Some key terminologies are discussed and highlighted according to our work. In the Section 3.2, the network deployment and cluster division of the entire network is discussed. The cluster sizes are kept dynamic and the whole process purely focuses on the efficiency and optimization of the entire network. Section 3.3 is about game formation. How can the work be aligned to a game model? How can the network players, strategies, and utility functions be defined? All similar questions are addressed and formulated. The key factors used in the work are formalized before being processed in the evolutionary game and evaluation process. These factors are discussed in the Section 3.4, while Section 3.5 covers the division of nodes, and Section 3.6 comprises the assessment by nodes. The list of parameters is mentioned. Those are used for the assessment of nodes. Also, an algorithm is applied in order to filter out the eligible nodes by BS. Section 3.7 explains the CH selection process, which is initiated by taking the sets of suitable, not suitable, and prohibited nodes. A CH selection game is discussed in detail and the last Section 3.8, covers the continual assessment by BS. The list of symbols is given in Table 2.

### 3.1. Preliminaries and Assumptions

In this work, a scenario of multi-hop data communication in a WSN is proposed, in which all the nodes are randomly deployed at finite distances with the central control node, the BS. The network is divided into clusters in such a way that there is no specific number of nodes per cluster. Each cluster can be identifiable by a unique ID as Cluster ID. The total number of clusters at time *T* can be represented by *KT*. Moreover, the nodes can switch to another cluster so the cluster sizes may be changed dynamically. The network can be defined in the terms of a game *G* as G={N, S, U}. Where, *N* represents the nodes, *S* denotes the strategy set for each node, and the utility function of each node can be denoted by *U*.

All the member nodes send the gathered data to their respective CHs. It is the duty of CHs to combine and summarize the observed collected data and forward it to the BS, either directly or through some other nodes. Other nodes can be CHs or CMs of the same or neighboring clusters. The network nodes are assumed to be rational to meet the requirement of game theory. The nodes are independent to either cooperate or not in the network. Moreover, the intelligent nodes can adjust their strategies to become CHs or get a better record for their participation level.

The fundamental routing mechanism is inspired by the dynamic source routing (DSR) [33] protocol. In DSR, the address of each involved node during the data transmission can be obtained from the received packets. The BS must be capable of investigating the level of involvement of all the network nodes by analyzing the received packets. This act is very essential in order for BS to enlist the selfish or noncooperative nodes in the cluster. A node can be said to be a selfish node when it does not cooperate with other nodes for some personal benefits.

A WSN with homogenous nodes has been proposed in this scheme. All the nodes have initially equal energy levels and their energy consumption ratios on various operations are also considered to be similar. For optimal accuracy and results, it is preferred to deploy BS near the center of the network. According to the basic principle of WSNs, the immediate nodes to the BS take more load as compared to other nodes. Therefore, it is desired to deploy nodes in such a way that nodes in each upper hop level are very dense as compared to the density of nodes at the lower hop level of the network. For example, the nodes directly connected with the BS, i.e., hop level 1, will be more densely deployed than the nodes lying in hop level 2.

### 3.2. Network Deployment and Cluster Division

Initially, when the nodes are deployed in the network all the nodes broadcast some control packets to share their parameters, such as energies and locations. The control messages are ultimately received at the base station, where major preliminary decisions are taken. The most important task for the base station is to divide the network nodes into clusters. The clusters are made according to the placement pattern of the nodes and consideration of some other important aspects. There are two tradeoffs while dividing the network into clusters for the sake of equalization of node energy consumption. (i) The nodes laying directly with the BS will have a higher load as compared to higher hop level nodes. The intermediate nodes will likely give the highest relaying service to all their backward nodes. (ii) Shorter routes can be obtained by dividing clusters of large sizes. In this case, the cost of intra-cluster communication will be very high, while making numerous small clusters produces reasonable costs of intra-cluster communication. However, in this case, the total relayed traffic is highly increased due to lengthy multi-hopped routes. So it is obvious that in smaller sized clusters the inter-cluster communication cost will be higher. Therefore, dividing the network into clusters cannot be considered as a simple procedure. Theoretically, we used a network consisting of 16 clusters, as given in Figure 1.

In this work, the cluster formation is based on the hop level and the number of nodes in each dividable region. The cluster sizes were kept moderate and the main objective was to get the optimal efficiency in intra-cluster and inter-cluster communication overhead. Moreover, the concept of logical clustering was used, in which the sizes of clusters can be unequal. There is a possibility of a dissimilar number of nodes at different segments in the network. At some places, the nodes may be densely deployed, while at another place nodes may be scattered when deployed. Therefore, by keeping these points the number of nodes is kept variable for each cluster in the network. Moreover, as the nodes operate in the network they consume their energies and the cluster sizes are also changed with the passage of time.

### 3.3. Game Formation

Initially, three classes of nodes were established i.e., suitable, not suitable, and prohibited nodes for the CH in each cluster. Two games were used to achieve the main objective of energy optimization and harvesting in the proposed work. The first game was mainly used for the selection of CHs. The mechanism consistently analyzes the performance of all the nodes in the network. In this process, an evolutionary game was used to assess both the suitable and unsuitable nodes for being cluster heads in their respective clusters. The effectiveness of each node in the clustering mechanism was compared. The second game gave an overall efficiency in terms of calculating various parameters in the network by equally assigning the tasks to the CHs and the CMs. It is a continuous process in which the games are invoked repeatedly. The entire mechanism can be considered as a single game consisting of these two games plus the initial class establishment phase, and these can be seen as the stages of the game. Additionally, we also incorporated a zero stage in the mechanism for the division of the network into clusters just after the deployment of nodes.

### 3.4. Matthematical Moel and Factor Formulization

Once clusters are formed then it is the responsibility of BS to nominate CHs in each cluster. The key parameters used are the remaining energies, the hop levels, and the degree of connectivity in terms of backward to forward node ratios, the density of nodes, and the selfishness level of nodes. These parameters are known as the factors in our work. The mathematical model needed to formulate these factors is developed in the following sections. We give detailed systems of equations for each factor individually.

#### 3.4.1. Energy Factors

Each node operates on a limited set of energy. Since all the nodes are supposed to sense, process, aggregate, send, receive, and forward data packets, they will consume energies with the passage of time. The initial energy level can be adjusted to 100 for each node.

The energy of node *i* can be denoted by $E_{i}$. The energy factor is associated with the energy cost of each node, which can be denoted as:(1)$$CostE_{i}=Cost_{i}\left(Sense\right)+Cost_{i}\left(Process\right)+Cost_{i}\left(agg\right)+Cost_{i}\left(Packet-Trans\right)$$
where $Cost_{i}\left(Packet-Trans\right)$ can be formed for *CHs* and *CMS* separately as:(2)$$Cost_{CH}\left(Packet-Trans\right)=Cost_{tx\left(CH,BS\right)}+Cost_{rx\left(CH,agg\right)}$$
where, $Cost_{tx\left(CH,\;BS\right)}$ is the cost in term of energy loss by sending a packet from *CH* to *BS*. $Cost_{rx\left(CH,agg\right)}$ is the energy cost of receiving a packet and aggregation from the *CMs*. The term $Cost_{tx\left(CH,\;BS\right)}$ can be further defined as:(3)$$Cost_{tx\left(CH,BS\right)}=d_{CH,BS}^{2}\cdot E\left(amp\right)+E\left(elec\right)$$
where, $d^{2}CH,\;BS$ is the physical distance between the *CH* and *BS*, while E(amp) is the transmission amplifier dissipation to get the needed signal strength, and E(elec) is the transmission circuitry dissipation. The cost of reception and aggregation can be defined as:(4)$$Cost_{rx\left(CH,agg\right)}=\overset{k}{\underset{CM=1}{\sum }}d_{CM}^{2}\cdot E\left(elec\right)+k\cdot E\left(agg\right)+E\left(lis\right)\;$$

In the above equation, we have a set of connected *CMs* denoted as *k*. *d_CM_* is the physical distance between the *CH* and the *CM*. E(lis) is the listening cost used by the antenna activation. On the other hand, the cost for the *CM* can be formalized as:(5)$$Cost_{CM}\left(Packet-Trans\right)=Cost_{tx\left(CM,CH\right)}=d_{CM,BS}^{2}\;+E\left(elec\right)$$

According to the above-defined energy model, we can conclude that the cost in terms of energy for a *CH* is higher than *CM*:(6)$$Cost_{CH}>Cost_{CM}$$

#### 3.4.2. Hop Level

It is obvious that some nodes in clusters cannot connect the BS directly or some may not be able to directly connect the CH of another cluster towards BS. The placement of nodes can also be considered for selecting CHs in clusters. In a single cluster, the nodes far away from the BS or other intermediate nodes cannot be assumed as suitable candidates for cluster headship. Therefore, the mechanism also incorporates the hop level procedure to determine the number of intermediate nodes in a route. Each node is assigned a hop level. The intermediate nodes with the BS are said to be hop level 1 nodes. Meanwhile, the nodes which are at a distance of one intermediate node from the BS can be said as hop level 2 nodes. Hop level of a node i can be denoted by $HpL_{i}$ with a condition $1\;\leq \;HpL_{i}\;\leq \;HpL^{max},$ where, $HpL^{max}$ is the hop level of last possible connected network boundary node.

#### 3.4.3. Degree of Connectivity

The nodes are randomly scattered in the network, therefore it is obvious that their degree of connectivity with each other will be different. The nodes’ connectivity or distance from the BS is primarily determined by their hop levels. Therefore, each node’s degree of connectivity can be accessed by taking its possible forward and backward nodes. The ratio of forward to backward nodes of node i can be calculated as:(7)$$fb=\frac{number\;of\;forward\;nodes+1}{number\;of\;backward\;nodes+1}.\;$$

It is possible that some nodes may directly connect the BS and some may have not any backward nodes. In order to avoid the undefined (divided by zero) error, one is added to both nominator and denominator in the equation.

#### 3.4.4. Density of Nodes

The importance of a node in the network can be determined by taking the number of nodes closely located with it. The densely deployed nodes usually sense similar data and forward to their CHs. Eliminating one or more nodes from a dense collection usually does not affect the regular operations of a network. For calculating the importance of a node, we must calculate the sets of closed neighbors (CNs). A set of CNs is the set of those nodes that are located very close to each other. The threshold distance for CNs can be set as ${\mathrm{CNDist}}^{\mathrm{Thres}}$. The following procedure can be used to calculate the distance among nodes:(8)$$d_{i,j}=\sqrt{{\left(x_{i\;}-x_{j}\right)}^{2}-{\left(y_{i\;}-y_{j}\right)}^{2}}.\;$$

$d_{i,j}$ can be used to calculate a simple distance between two nodes, *i* and *j*. The propagation model $P\left(d_{i,j}\right)$ can be defined to get the set of closed neighbors as:(9)$$P\left(d_{i,j}\right)=C_{t}\frac{P_{t}}{d^{4}}.\;$$

In the above equation, the constant $C_{t}$ denotes the transceiver characteristic, and $P_{t}$ the transmitting power of two connected nodes. A set of CNs can be calculated for node *i* as:(10)$$CN_{i}=\left\{j:P\left(d_{i,j}\right)\leq CNDist^{Thres}\right\}.\;$$

While manipulating the nodes for CH selection, it is important to consider the importance of each node in the cluster. For calculating the importance of a node we can consider its CNs with their energy levels. A node having more CNs with higher energies will be considered as more suitable for being CH. The importance of node *i* is denoted by $\lambda_{i}$ in the following equation.
(11)$$\lambda_{i}=\;\{\begin{array}{c} \;\frac{\sum_{j=1}^{cn}E_{CNj}^{t}}{cn}\;cn\neq 0\\ \;1\;cn=0 \end{array}$$

The value of $\lambda_{i}$ can be obtained by taking the number of all CNs denoted as cn and the sum of all CNs energies.

#### 3.4.5. Selfishness

The network uses a source routing mechanism, therefore, the participation level of each node in the network can be easily calculated at the BS by checking the headers of received packets. In source routing the identities of all the involved nodes in a transmission are recorded in the header therefore it is easy to determine the nodes participation in transmission. The nodes can be marked as normal or selfish nodes by checking the values of their participation in the network. A threshold level of participation $Part^{Thres}$ is used to determine the participation level of nodes.
(12)$$p_{CM}=\frac{TSS_{CM}}{\sum_{j=1}^{cn}\left[TSS_{j}\right]/cn}$$
where, $p_{CM}$ denotes the participation value of each *CM*, which can be calculated by taking the total number of sessions sent by *CM*, i.e., $TSS_{CM}$ to CH and the average of the *TSS* sent by its *CNs*. If a node has $p_{CM}<\;Part^{Thres}$, it is considered as a selfish node. Such nodes cannot be considered for the candidacy of cluster headship.

### 3.5. Division of Nodes

In the game, we always assume that the players and nodes are intelligent and can select their own strategies. In this theory, it is also notable that each player should know about the preferred strategies of its neighborhood players. A game theoretic approach is used in an evolutionary pattern to select CHs for all the clusters in the network. The players can be divided into three segments according to their parameters. The network can be denoted by N having three types of nodes, i.e., *N* = {α, β, PCH}. Where, α are the nodes suitable for cluster-headship, while β are the nodes that are not suitable for the CH positions. However, it is not necessarily the case that β cannot be made CH. For optimization and during the CH evaluation, the BS may consider these nodes for cluster-headship. However, for optimal results α nodes are always considered as the most suitable node to be CHs. Optimal results are those results that we desire to get from the CHs using game theoretic modelling. PCH are the nodes that are considered to be prohibited from the position of CH. Such nodes may have a noncooperative history or have an extraordinary location in the network, which does not allow them to become CHs.

### 3.6. Assessment by Nodes

Initially, the nodes and the BS design a borderline to distinguish all the α and β nodes, while taking all the possible parameters and strategies of the nodes. According to Table 3, the preferences are set as:

A threshold value TE (threshold for eligibility) is set by the BS to filter out the eligible nodes while processing the Algorithm 1.


**Algorithm 1: Nodes filtration for eligibility**
1. $FOR\;EACH\;Cluster_{ID}\;in\;N\;$
2. $\;\;\;FOR\;EACH\;node\;i\;in\;Cluster_{ID}$
3.
$\;\;\;\;\;\;\;\;\;\;\;\;\;\;\;\;\;IF\;p_{i}\;<\;Part^{Thres\;}then\;$
4.
               add i to PCH
5.    ELSE
6.
$\;\;\;\;\;\;\;\;\;\;\;\;\;\;\;\;\;\;Calculate\;CANDIDANCY_{i}=(E_{i}+$
$\;HpL_{i}\;+\lambda_{i})*\left(\frac{1}{fb}\right)$
7.
$\;\;\;\;\;\;\;\;\;\;\;\;\;\;\;\;\;\;\;\;\;\;IF\;CANDIDANCY_{i}>TE\;THEN\;$
8.
                           add i to α SET
9.      ELSE
10.      *add i to β SET*11.     END IF
12.     END IF
13.     
END FOR
14.    
END FOR


TE is smartly adjusted for each round of CH selection by the BS. The value of TE declines with the passage of time as the nodes exhaust their energies and ultimately die in the network.

Algorithm 1 provides the optimal solution because each network node, either CH or CM, is accessed only once to determine its class by checking all its possible parameters. The nodes are scanned only once by using a nested loop, therefore, we can say that Algorithm 1 can be executed optimally at maximum of O(n) times. Where, n is the total number of nodes in the network. Moreover, in Algorithm 1 each cluster is taken and then all the nodes inside that cluster are classified for α, β, and PCH. The ultimate result of this Algorithm 1 enables us to determine the suitability of nodes for being CHs.

### 3.7. Cluster Head Selection

Since the nodes are rational, they can adapt their own strategies to either become CHs or not. The strategy set of node *i* can be $S_{i}=\left\{CH,\right\}$ CM. The strategies align the direction of nodes towards their future responsibilities of either being a CH or a CM. Additionally, the parameters also influence the possibilities in this regard. Each node is continuously treated by the BS with a cardinal value V, which could be either profit (Π) or penalty (π). Each time, a value is given to the target node by the BS to adjust its likeness for either α or β. The utility function for each node to become either of CH or CM is directly associated with the value type of V assigned by the BS. The main utility function $U_{i}$ of each node *i* can be defined as:(13)$$U_{i}=\Pi_{i}-\pi_{i}.\;$$

The main motivation behind the utility function is to allow the nodes of CM status to have α capability. A node having higher Π than its π values will be a node having positive utility. Such nodes will have a strong chance of becoming CHs during the cluster selection process.

The division of profit and penalty is also further divided into the classes according to following Table 4:

The payoff matrix of nodes having all the possibilities can be seen in Table 5.

**Lemma** **1.**
*According to the second stage of our game, the selection for CH exists for α nodes and the selection of CM exists for β nodes at a certain point of time in the network.*


**Proof.** Suppose the possibility of an α node to become a CH is *x*. So we can say that according to this assumption, an *α* node must have (1 − *x*) possibility for being a *CM*. Similarly, for taking the nodes of the *β* set for *CH* and *CM* we can have y and (1 − *y*). The utility functions for selecting the *CH* and *CM* are as in the following:

(14)$$U_{\alpha \left(CH\right)}=\;y\;\Pi \;+\;\left(1-y\right)\left(\;\Pi \;+2\;\pi \right)\;$$

(15)$$U_{\alpha \left(CH\right)}=\;y\;\Pi \;+\;\Pi +\;2\;\pi -y\Pi \;-2y\;\pi$$

(16)$$U_{\alpha \left(CH\right)}=\;\Pi \;+\;2\;\pi -2y\;\pi$$

(17)$$U_{\alpha \left(CM\right)}=\;y\left(\Pi \;-\;\pi \right)\;+\;\left(1-y\right)\left(-\;\pi \right)$$

(18)$$U_{\alpha \left(CM\right)}=\;y\;\Pi \;-\;\pi$$

Equation (16) defines the expected utility function $U_{\alpha \left(CH\right)}$ of α nodes for being selected as *CH*, while the utility for being selected as *CM* of the same class α is defined by $U_{\alpha \left(CM\right)}$ in Equation (18).

The average earning of a node can be denoted by Equation (20) as:(19)$${\bar{U}}_{\alpha }=x\left(\Pi \;+2\;\pi \;2y\;\pi \right)+\left(1-x\right)\left(y\;\Pi \;-\;\pi \right)\;$$

(20)$${\bar{U}}_{\alpha }=x\;\Pi \;+3x\;\pi \;-2xy\;\pi \;+y\;\Pi \;-xy\;\Pi \;-\;\pi$$

In evolutionary games, the replicator dynamic equation is considered as the most important aspect. The stability and convergence of evolutionary stability and Nash equilibrium can be summarized by this dynamic analysis. Equation (24) shows the replicator dynamic analysis of node α as below:(21)$$\frac{dx}{dt}=F\left(x\right)\;$$

(22)$$F\left(x\right)=\;x\left(U_{\alpha \left(CH\right)}-{\bar{U}}_{\alpha }\right)$$

(23)F(x)=x(Π +2 π−2y π−x Π −3x π+2xy π −y Π +xy Π + π) 

(24)F(x)=x(1−x)[ Π +3 π−y(2 π− Π)].

Similarly, the expected utility function for non-suitable nodes, β, for both cases of being selected as CHs or CMs can be defined by the following equations.

(25)$$U_{\beta \left(CH\right)}=x\;\Pi \;+\left(1-x\right)\left(\Pi \;+\;\pi \right)=\;\Pi \;+\;\pi \;-x\;\pi$$

(26)$$U_{\beta \left(CM\right)}=x\left(\Pi \;+2\;\pi \right)+\left(1-x\right)\left(-\;\pi \right)=\;\Pi x+3x\;\pi \;-\;\pi ,$$

The average revenue obtained by each β node is as follows:(27)$${\bar{U}}_{\beta \;}=y(\Pi +\pi -x\;\pi )+(1-y)(\Pi \;x+3x\;\pi -\pi )$$

(28)$${\bar{U}}_{\beta }\;=\;y\;\Pi \;+\;2y\;\pi \;-\;4xy\;\pi +\Pi \;x\;+\;3x\;\pi \;-\;\pi \;-\;\Pi \;xy.$$

The replicator dynamic equation for each node in the β class can be similarly obtained as:(29)$$\frac{dy}{dt}=G\left(y\right)\;$$

(30)$$G\left(y\right)=y\left(U_{\beta \left(CH\right)}-{\bar{U}}_{\beta }\right)\;$$

(31)G(y) = y(Π + π − x π −  Π y − 2y π + 4xy π − Π x − 3x π + π + Π xy)

(32)G(y)=y(1−y)[ Π +2 π −(+4 π)x]

Equation (24) indicates that $x^{*}=0$
$x^{*}=1$ can be obtained for α class of nodes as stable strategies sets. As per the flow, the values of Π and π cannot be negative, therefore Π + 3π − *y*(2 π − Π) will always return a positive value. It shows that the evolutionary settlement of nodes from classes α and β for being CHs or CMs can be formed.

In the initial phase, the BS classifies nodes to be either α or β, based on their suitability for the cluster headship in their clusters. The utility function for both classes can be calculated by the previous equations already discussed in this subsection. All the nodes are intelligent and can adjust their performance in the network towards the candidateship of either CH or CM. The BS continuously tracks their performance and benefits them accordingly. The nodes that higher revenues are always selected as CHs. The revenue is controlled by the BS while analyzing their suitability in the network. The BS continuously assesses the performance of CHs and designs the revenue for each node to direct them towards the responsibility of CH. The following Figure 2 shows a clear view of our scheme.

### 3.8. Continual Assessment by the BS

The CHs are always switched with the passage of time due to their degrading performance, caused by either their own energy or by other nodes. The energies and presence of neighboring nodes can also affect the performance of a CH. It is always desired to select the most efficient and economical CHs in each cluster. In our work, the CHs are selected through the evolutionary game, as discussed in the previous section. However, it is the responsibility of BS to keep an eye on the performance of all the CHs throughout the network lifetime. A multi-factor approach similar to a Stackelberg game is designed to evaluate the condition of CHs and CMs in each cluster by the BS. The BS always keeps a list of leading CHs, along with their following CHs. The leading nodes are considered as the most suitable nodes for cluster headship. Once a leader selected as CH, it is also evaluated again and again and may be switched back to CM by comparing it with the leaders. The key parameters assessed by the BS are: Residual energy, number of closed neighbors, number of neighbors, location of a node, channel quality, and throughput. The residual energy can be seen as the available energy of nodes, which can be used for processing, sensing, and transmission. The closed neighbors can have an adequate impact on the performance on the CH and responsibilities among them can be divided or shared. The number of neighbors can be used to determine cluster size. The higher number of neighbors will lead to a higher level of dependence of CMs or a single CH. In such cases, it is obvious that the CH will exhaust its energy comparatively quickly. The location of the node can also affect the level of energy dissipation on the communication with neighbors. The BS also check the throughputs from different CHs and analyzes their performance according to their inputs in the network. These factors can be represented as:(33)$$EVAL=\{Ev_{1},Ev_{2},Ev_{3},Ev_{4},Ev_{5},Ev_{6}\}\;$$

(34)$$EVAL=\left\{\begin{array}{c} Energy,\;Location,\;Neighbors,\;\\ \;\lambda i,\;Channel,\;throughput \end{array}\right\}.$$

The parameters are then graded into different levels to check their weightage and compared with each other. The levels are as follows:(35)LEVELS={1,2,3,…,10}. 

Level 1 indicates the worst case or the least value for a grade, while level 10 represents the best case. The nodes that have an aggregate of highest levels for each of their evaluation parameters are considered as the best suitable candidates for cluster-headship. Such nodes are given additional benefits by the BS to make their position stronger for the next rounds of CH selection.

It is obvious that the BS cannot investigate all the nodes for these values of parameters periodically. Such types of information exchange can put extra communication overhead on nodes and can reduce the network performance. A probabilistic model is used to determine the energy and $\lambda_{i}$ values for each node in the network. The DSR protocol is used for routing in the network, therefore, it is easy to extract the IDs of all the involved nodes in a communication session. The BS can easily estimate the energy consumption of each involved node once the list of these nodes and the amount of transmitted data is obtained.

## 4. Simulation Results

A plane domain of the area of 500 × 500 is selected to deploy sensor nodes in a random fashion. The list of parameters used is given in the Table 6. This work is cluster-based, therefore, the performance metrics are compared with LEACH and LEACH-C [16]. Moreover, this work mainly depends the evolutionary game, therefore, GEEC [30] and GTEB [31] are also taken for comparative studies. Our game theoretic solution for power management in wireless sensor networks is referred to as GTSPM.

This work and other taken protocols were tested and compared. The results indicate that the GTSPM protocol produces comparatively better qualitative results as compared to the tested protocols. The key performance metrics in this section are (a) the number of dead nodes, (b) average energy consumption, and (c) average throughput and the number of nodes and the pause times.

Figure 3 shows the number of alive nodes recorded at 100 intervals. In total, 300 nodes were taken in an area of 1000 × 1000 m^2^. All the protocols were performing well until the 50 time pause point. The results show that the performance of LEACH and LEACH-C are comparatively low. The number of dead nodes in the GTSPM protocol, over the entire time, is lower than all the compared protocols. This is because multiple options for energy efficiency in the network are considered. The key improvement is obtained by considering the CNs and continuous evaluation of CHs, combined with an evolutionary game model.

Figure 4 shows the analysis of all protocols with respect to the number of alive nodes against the varying number of nodes. The values were recorded at 90 pause times, for each set of nodes, in connection with the previous results. It is clear that GTSPM gives a much better performance when the network size is increased. This is because with a higher number of nodes, the number of CNs is also increased. It is possible that the higher number of CN sets suggest many suitable CHs during the selection process.

The results for energy consumption can be seen in Figure 5. In this experiment, 300 nodes were taken and we can see that the evolutionary models GEEC, GTEB, and GTSPM performed well, as compared to LEACH and LEACH-C. We can conclude that energy management is much better in evolutionary game-based protocols. Due to additional approaches, our protocol, GTSPM, is slightly better than other protocols after time pause 70. The energy efficiency in GTSPM is also additionally added by constructing the set of prohibited nodes from cluster headship i.e., PCH.

Another energy consumption analysis against the varying number of nodes is given in Figure 6. The values were recorded at time pause 50 for each set of nodes. GTSPM gave similar results as those of GTEB. However, it is notable that the performance of GTSPM is associated with the network size. Other protocols did not respond well to the increased number of nodes, as compared to GTSPM. It can be noted that at the beginning, with fewer nodes, the performance of GTSPM was lower than GTEB. As the number of nodes increased, the average remaining energy of all the nodes in GTSPM protocols rose higher than GTEB. This was because of additional mechanisms introduced by GTSPM.

The performance metric for throughput can be seen in the Figure 7 and Figure 8. Figure 7 shows the results for 300 nodes in an area of 1000 × 1000 m^2^ over 100 time intervals. Prohibiting the selfish nodes for the CH selection process and continuous evaluation of CHs by the BS caused the improvement in throughput as compared to other evolutionary clustering algorithms. In this metric GTSPM outperformed all the protocols with slightly higher margins. Similarly, in Figure 8, the average throughput is shown for the various numbers of nodes. The results were recorded at time pause 90 in connection with the experiments in Figure 7. In this experiment, all the protocols gave a positive response to the increased number of nodes in the network. GTSPM gave similar better results in this case too.

## 5. Conclusions and Future Work

Various evolutionary game-based approaches have been proposed to overcome the cluster division and CH selection issues in WSNs. The main motivation of such mechanisms is to reduce the energy consumption by balancing load among the nodes and select the most appropriate CHs. In this work, we have optimized the energy efficiency by incorporating some additional procedures, integrated into the evolutionary game. Initially, the clusters are divided intelligently so that each cluster is formed in its optimal position. Then, to initiate the game, some key parameters are processed to form three different sets of nodes; suitable, non-suitable, and prohibited for CH. The nodes’ participation level is considered to mark the prohibited nodes. For the suitability of nodes, the energy, hop level, density, and degree of connectivity of all the nodes are considered. The evolutionary game takes both suitable and non-suitable nodes, along with incentives granted by the BS. The BS continuously monitors the network by using various nodes’ and CHs’ related parameters and grants incentives in the form of profit and penalties. The work is compared with some fundamental and state-of-the-art existing protocols. The analysis results indicate the excellence of our work by giving adequately better values as compared to other protocols. In future, we are planning to implement and validate our proposed mechanism using an experimental testbed of 100 nodes. Using this testbed, we will be able to get real world results using various network setups and more extensive simulations based on parameter variations.

The heterogeneity of nodes can also be addressed and incorporated in the proposed system. The base protocol, DSR, can be replaced by some other protocols, along with some additional procedures. A Stackelberg game can be incorporated in the proposed mechanism by introducing the sets of leading and following nodes.

## Figures and Tables

**Figure 1 sensors-19-03835-f001:**
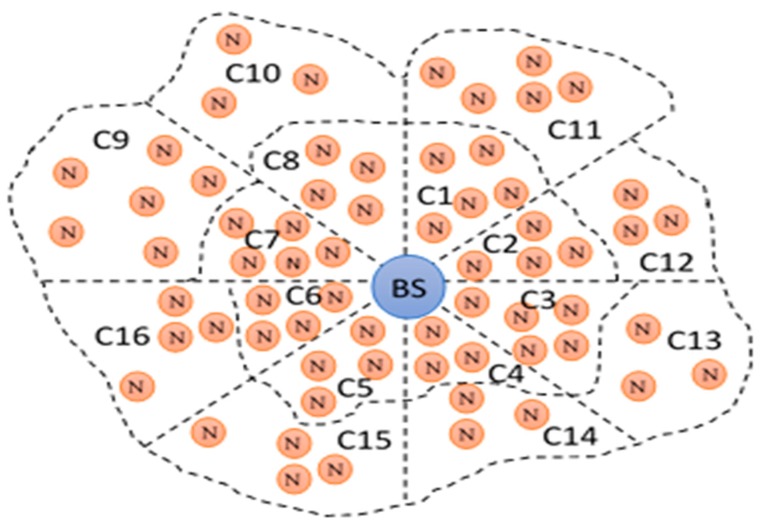
Network divided into 16 clusters.

**Figure 2 sensors-19-03835-f002:**
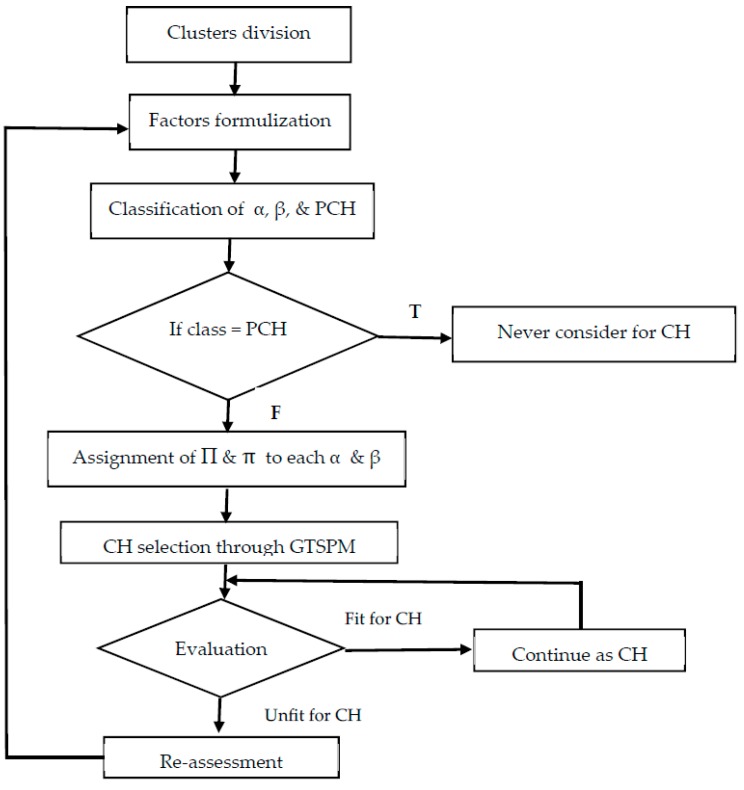
Flowchart to clearly show the proposed scheme.

**Figure 3 sensors-19-03835-f003:**
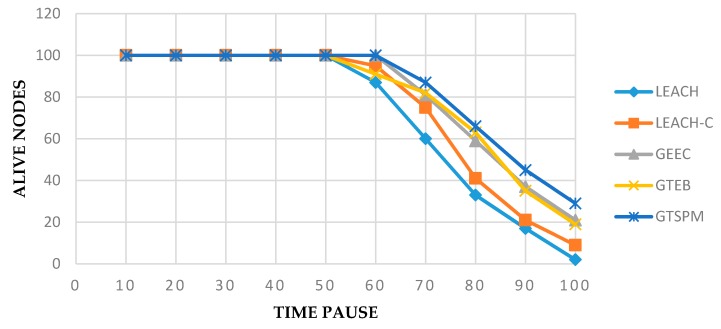
Number of alive nodes in different time pauses.

**Figure 4 sensors-19-03835-f004:**
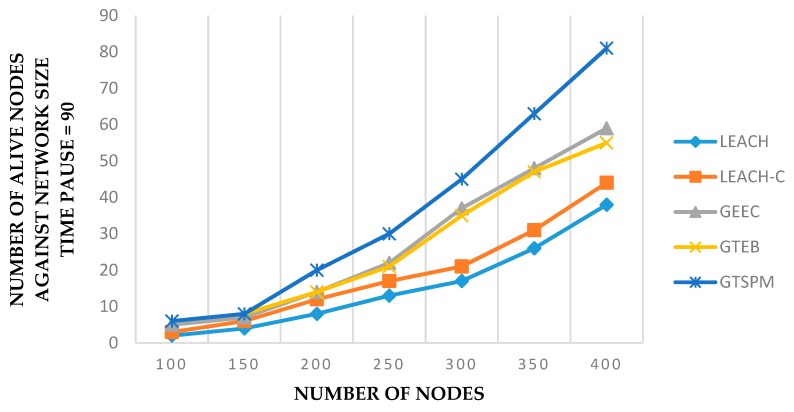
Number of alive nodes with varying number of nodes (pause time = 90).

**Figure 5 sensors-19-03835-f005:**
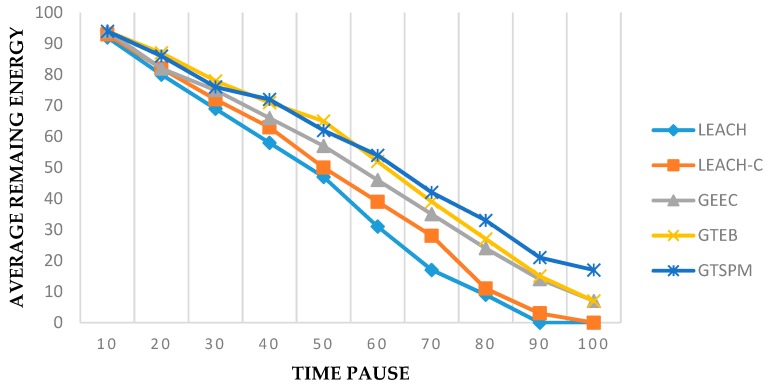
Average energy consumed over time (total nodes = 300).

**Figure 6 sensors-19-03835-f006:**
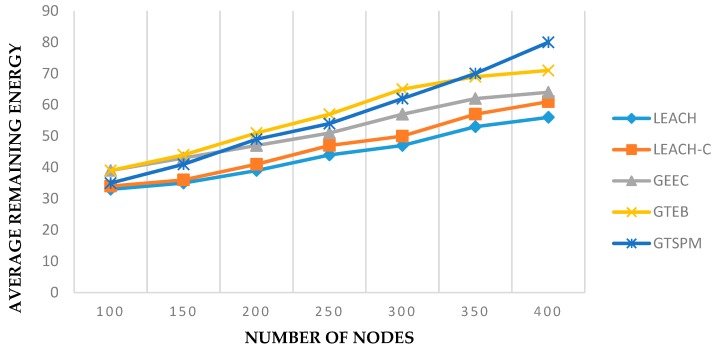
Average energy consumed with varying network size (pause time = 50).

**Figure 7 sensors-19-03835-f007:**
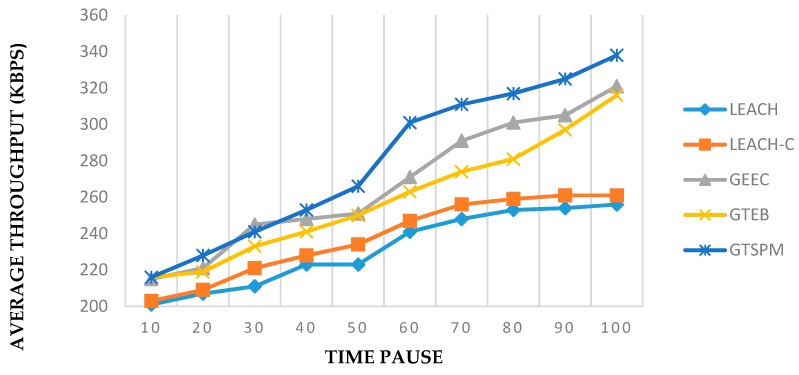
Throughput with time pauses (total nodes = 300).

**Figure 8 sensors-19-03835-f008:**
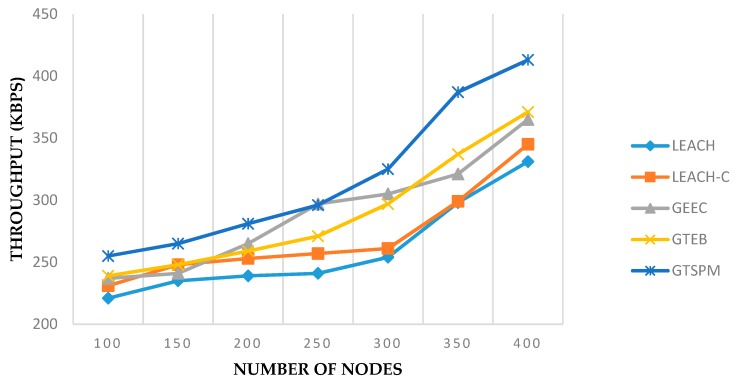
Throughput with varying network size (pause time = 90).

**Table 1 sensors-19-03835-t001:** Comparison of advantages and disadvantages of related work.

Method	Key Feature	Advantages	Disadvantages
Distributed clustering with data fusion [10,11]	Multi-hop and intra-cluster communication	The need for centralized stations for cluster CH formation is eliminated	Selfishness of nodes is ignored.
Cognitive radio WSN (CR-WSN) [12,13,14,15]	Utilization of increased spectrum	Extends WSNs lifetime	Clustering in CR-WSNs has not yet been matured.
Energy conservation via efficient routing algorithms like LEACH, LEACH-C [16], PEGASIS [17,18]	Nodes of clusters made rich in information about neighboring nodes	Extend WSNs’ lifetime	Selfishness of the node is ignored.
One Hop Cluster-Head Algorithm (OHCH) [19].	CH selection	Prolonging the network lifetime and reducing the network data latency	Ignoring the distance factor as nodes get away and deplete energy quickly
Dynamic network state learning model (NSLM) [20]	Hidden Markov model (HMM) and Lagrange multiplier-based approach	Outperformed in terms of buffer cost, holding cost, overflow, energy consumption, and bandwidth usage	Other optimization approaches are ignored to compare.
Noncooperative game theoretic approach in clustering [22]	Dependability assessment mechanism for heterogeneous WSNs	Reliability and availability measures for susceptible sensor nodes improved	Not all possible security measures considered.
Coalitional game theory [24]	Based on topological structure	Extend WSNs’ lifetime via finding the cheapest route	How to choose corresponding leaders is not mentioned in the work.
Bayesian game [25]	Bayesian game to form static game	Extend WSNs’ lifetime	Imperfect information.
Game theory [28]	A trade-off between energy conservation and network throughput, double-time Nash equilibrium	Balances energy consumption	Selfishness of nodes are ignored.
Game theory-based energy efficient clustering routing protocol (GEEC) [30]	The proposed mechanism is compared with LEACH and LEACH-C	Extend WSNs’ lifetime	No guarantee of the connectivity and robustness of the network.
An evolutionary game [31]	Combined the evolutionary game with the classical GT.	Extend WSNs’ lifetime and provides a better delivery ratio.	Selfishness of nodes are ignored.

**Table 2 sensors-19-03835-t002:** List of Symbols.

S#	Symbol	Description
1.	CH	Cluster head
2.	*Cluster_ID_*	Cluster with an identity number
3.	K^T^	Total number of clusters at time T
4.	G	Game model
5.	N	Nodes/number of nodes
6.	S	Strategies
7.	U	Utility function
8.	CM	Cluster member
9.	$CostE_{i}$	Energy cost of node *i*
10.	$Cost_{i}\left(Sense\right)$	Cost of sensing
11.	$Cost_{i}\left(Process\right)$	Cost of processing data
12.	$Cost_{i}\left(agg\right)$	Cost of aggregation of data
13.	$Cost_{i}\left(Packet-Trans\right)$	Cost of a single packet transmission
14.	$Cost_{CH}$	Energy cost of cluster head
15.	$Cost_{CM}$	Energy cost of cluster member
16.	$HpL_{i}$	Hop level of node *i*
17.	$HpL^{max}$	Maximum hop levels in the network
18.	fb	The ratio of forward nodes to backward nodes
19.	$d_{i,j}$	Distance between nodes *i* and *j*
20.	*CN*	Closed neighbors
21.	$CNDist^{Thres}$	Threshold distance to determine CNs set
22.	$P\left(d_{i,j}\right)$	Propagation model for the distance between nodes *i* and *j*
23.	$C_{t}$	Transceiver characteristics
24.	$P_{t}$	Transmission power
25.	$\lambda_{i}$	Importance of node *i*
26.	cn	Number of CNs
27.	$Part^{Thres}$	The threshold value for participation
28.	$p_{CM}$	The participation level of cluster member
29.	$TSS_{CM}$	Total Session Sent by the cluster member
30.	A	Set of nodes suitable for CH
31.	Β	Set of nodes not suitable for CH
32.	PCH	Set of nodes prohibited for CH
33.	TE	Threshold value for eligibility of being CH
34.	$\Pi_{i}$	Profit value assigned to the node by BS
35.	$\pi_{i}$	Penalty value assigned to the node by BS
36.	${\bar{U}}_{\alpha }$	Average profit earning of a node
37.	${\bar{U}}_{\beta }$	The average penalty for a node
38.	$x^{*}$	Stable strategy set
39.	EVAL	Parameters set for evaluation

**Table 3 sensors-19-03835-t003:** Considerations for cluster head (CH) selection.

S#	Factor	Impact
1.	Remaining Energy	A higher level of remaining energy leads to a higher possibility of being α
2.	Selfishness	A node being declared as selfish must be put in PCH
3.	Hop Level	Nodes nearer to the BS are considered more suitable for α set
4.	Density	A node having many closed neighbors is preferred as an α
5.	Degree of Connectivity	A node having fb ratio nearer to 1 is suitable for α

**Table 4 sensors-19-03835-t004:** The further division of profit and penalty.

Node Class	Profit (Π)	Penalty (π)
α	Π (α)	π(α)
β	Π (β)	π(β)

**Table 5 sensors-19-03835-t005:** The payoff matrix of nodes to be CH.

α	β	
	CM	CH
CM	(−π, −π)	(Π − π, Π)
CH	(Π + 2 π, π)	(Π, Π)

**Table 6 sensors-19-03835-t006:** List of parameters.

Parameter	Value
Simulation Environment	MATLAB under Windows
Area	500 × 500/1000 × 1000 m^2^
Network Type	WSN/Cluster-based
BS Location	Center (250,250/500,500)
Number of Nodes	100 to 400
Node Distribution	Random
Comparisons	LEACH, LEACH-C, GEEC, GTEB
Initial Energy	100 J
Rx Power	0.3 J
Tx Power	0.6 J
Movement Trace	Off
Cluster Size (GEEC/GTEB/GTSPM)	Game-based (Varying)
Cluster Size (LEACH, LEACH-C)	9 nodes
Traffic Source	CBR
Packet Protocol	TCP
$CNDist^{Thres}$(GTSPM)	100 m

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
