# Peer review of "Game Theoretic Solution for Power Management in IoT-Based Wireless Sensor Networks"

_sensors, 2019, doi:10.3390/s19183835_

Round 1

Reviewer 1 Report

In general, the studied topic is very timely and the ideas are interesting. However, there are some parts of the paper requiring justifications. First, the contributions of the paper are not clear compared to existing works in the literature. Second, the authors should discuss the assumptions adopted in the paper. Third, the authors should improve the literature survey. As a result, the reviewer would suggest a major revision.

1) since the problem is related to 5G, the following book may provide you with a good reference on 5G related topics:

V. W.S. Wong, et al, “ Key Technologies
for 5G Wireless Systems:?. Cambridge University Press, 2017.

2) The contributions of the paper are not clear compared to existing works in the literature.
3) If possible, please formulate the considered problem mathematically.
See

D. W. K. Ng, E. S. Lo and R. Schober, "Robust Beamforming for Secure Communication in Systems With Wireless Information and Power Transfer," in IEEE Transactions on Wireless Communications, vol. 13, no. 8, pp. 4599-4615, Aug. 2014.

4) The authors should discuss the optimality of the proposed algorithm.
5) What is the computational complexity of the proposed algorithm?

Author Response

Reviewer 1

S.No

Reviwer1 comment

Our reply

1

Since the problem is related to 5G, the following book may provide you with a good reference on 5G related topics:

V. W.S. Wong, et al, “ Key Technologies
for 5G Wireless Systems:?. Cambridge University Press, 2017

The book is referred in section 1 line no 35.

2

The contributions of the paper are not clear compared to existing works in the literature. 

Discussed now in line no 185-192.

Although some of the abovementioned algorithms and protocols based on game theory can achieve network topology control and improve network performance, they cannot guarantee the connectivity and robustness of the network. Additionally, the parameters like residual energy, degree of connectivity, distance, and energy efficiency of the nodes are not fully and accurately considered. Nor these parameters are taken combined and rationalized as a whole. Therefore, we need a mechanism to consider the proper selection of CH by considering appropriate parameters rationally, thus we can manage the power efficiently. Our proposed scheme contributed better results as given in section 4.

3

The authors should improve the literature survey

Typos are removed and comparison table is added see table 1.

3

If possible, please formulate the considered problem mathematically. 
See

D. W. K. Ng, E. S. Lo and R. Schober, "Robust Beamforming for Secure Communication in Systems With Wireless Information and Power Transfer," in IEEE Transactions on Wireless Communications, vol. 13, no. 8, pp. 4599-4615, Aug. 2014.

Mathematical model and formulization section 3.4 line no 294.

4

The authors should discuss the optimality of the proposed algorithm.

Discussed line no 429.

The algorithm1 provides optimal solution because each network node, either CH or CM, is accessed only once to determine its class by checking its all possible parameters. Since the nodes are scanned only once by using a nested loop, therefore we can say that the algorithm1 can be executed optimally at maximum of O(n) times. Where n is the total number of nodes in the network. Moreover in algorithm1 each cluster is taken and then all the nodes inside that cluster are classified for  β and PCH. The ultimate result of this algorithm1 enables us to determine the suitability of nodes for being CHs.

5

What is the computational complexity of the proposed algorithm?

Addressed at line no 429-435.

The algorithm1 provides optimal solution because each network node, either CH or CM, is accessed only once to determine its class by checking its all possible parameters. Since the nodes are scanned only once by using a nested loop, therefore we can say that the algorithm1 can be executed optimally at maximum of O(n) times. Where n is the total number of nodes in the network. Moreover in algorithm1 each cluster is taken and then all the nodes inside that cluster are classified for  β and PCH. The ultimate result of this algorithm1 enables us to determine the suitability of nodes for being CHs.

Reviewer 2 Report

Correct an elaborated article dealing with dynamic game theory in sensor networks. Unfortunately authors did not go into direction various network setups and more extensive simulations based on parameter variations (e.g. decrease of energy over time and reconfiguration of network) but still this is article that deserves to be published. Also, pseudo code is strange at one place because it has two identical instructions. 

Author Response

S.No

Reviewer 2 comment

Our Response

1

Unfortunately authors did not go into direction various network setups and more extensive simulations based on parameter variations (e.g. decrease of energy over time and reconfiguration of network) but still this is article that deserves to be published.

   Thank you for the valuable comment. We are planning to address these comments in future where we will use “leader and follower” game theoretic model with different combinations and variations of parameters as discussed in section 5 (Conclusion and future work.)

2

Pseudo code is strange at one place because it has two identical instructions. 

Pseudo code is corrected, line no 6 in algorithm1 is not repeated now.

Reviewer 3 Report

In this paper, an evolutionary game approach was proposed to select cluster head (CH) of wireless sensor networks. It is a meaningful research to apply game theory for CH selection. However, this paper is not well prepared and presented. The following comments may help authors to improve the paper:

In abstract, lines 23-24, "the existing mechanisms do not consider many possible parameters associated with the smart nodes in WSNs." which parameters? Line 25, "some vital parameters". which parameters? Reference [1] was published by authors. This paper is not suitable to be used as a reference for applications of WSNS. The introduction and related work are long. However, authors didn't clearly mention current research gap and contribution of this research. Several related papers were listed. What are the advantages of these methods? What are the disadvantages? How this paper to address the current research gap? Use a flowchart to clearly illustrate the proposed method. Line 430, "for best optimal results". What is "best optimal"? In table 5, unit of Energy is needed. The simulations were compared with other published results. How to validate simulation results given in this paper? Lot of typos in the paper. 

In summary, this paper should be significantly revised and improved before considering for acceptance.

Author Response

S.No

Reviwer3 comment

Our Response

1

In abstract, lines 23-24, "the existing mechanisms do not consider many possible parameters associated with the smart nodes in WSNs." which parameters?

The abstract is rearranged and the comments are addressed.

2

Line 25, "some vital parameters". Which parameters?

The parameters are added.

3

Reference [1] was published by authors. This paper is not suitable to be used as a reference for applications of WSNS.

The reference is removed.

4

The introduction and related work are long. However, authors didn't clearly mention current research gap and contribution of this research.

We added a table no 1 for clarification and the a complete  paragraph above the table to clarify research gap and our contribution

5

Several related papers were listed. What are the advantages of these methods? What are the disadvantages?

Table 1 is added 

6

How this paper to address the current research gap?

Discussed at line no 185-192.

Complete  paragraph above the table1 is added to address current research gap

7

Use a flowchart to clearly illustrate the proposed method

Flowchart is added as figure 2.

8

Line 430, "for best optimal results". What is "best optimal"?

The paragraph is rephrased for clarity at line no 411-413.

However, for optimal results α nodes are always considered as the most suitable node to be CHs. Optimal results are those results which we desire to get from the CHs using game theoretic modelling.

9

In table 5, unit of Energy is needed

Corrected now in Table 6

10

The simulations were compared with other published results. How to validate simulation results given in this paper?

The comment is addressed now in section 5 line no 657-662.

Our proposed mechanism is compare with state of the art existing protocols. The work is compared with some fundamental and state of the art existing protocols. The analysis results indicate the excellence of our work by giving adequately better values as compared to other protocols. In future we are planning to implement and validate our proposed mechanism using experimental testbed of 100 nodes. Using this testbed we will be able to get real world results.

11

Lot of typos in the paper

Proof reading has been done

Round 2

Reviewer 3 Report

Authors carefully addressed my comments and concerns except the simulation results validation. How readers know the results shown in the paper are correct? Since authors published a related work at IEEE Access, the reviewer believes that authors have verified simulation results. Therefore, the reviewer recommends to accept this paper for publication.